# Hierarchical Perceptual and Predictive Analogy-Inference Network for Abstract Visual Reasoning

## ABSTRACT

Advances in computer vision research enable human-like high-dimensional perceptual induction over analogical visual reasoning problems, such as Raven's Progressive Matrices (RPMs). In this paper, we propose a Hierarchical Perception and Predictive Analogy-Inference network (HP$^2$AI), consisting of three major components that tackle key challenges of RPM problems. Firstly, in view of the limited receptive fields of shallow networks in most existing RPM solvers, a perceptual encoder is proposed, consisting of a series of hierarchically coupled Patch Attention and Local Context (PALC) blocks, which could capture local attributes at early stages and capture the global panel layout at deep stages. Secondly, most methods seek for object-level similarities to map the context images directly to the answer image, while failing to extract the underlying analogies. The proposed reasoning module, Predictive Analogy-Inference (PredAI), consists of a set of Analogy-Inference Blocks (AIBs) to model and exploit the inherent analogical reasoning rules instead of object similarity. Lastly, the Squeeze-and-Excitation Channel-wise Attention (SECA) in the proposed PredAI discriminates essential attributes and analogies from irrelevant ones. Extensive experiments over four benchmark RPM datasets show that the proposed HP$^2$AI achieves significant performance gains over all the state-of-the-art methods consistently on all four datasets.

## CCS CONCEPTS

• **Computing methodologies** → **Cognitive science**; **Image representations**; **Scene understanding**.

## KEYWORDS

Analogical Visual Reasoning, Raven's Progressive Matrix, Intelligence Quotient Test, Transformer, Predicting-and-Verifying

## 1 INTRODUCTION

Analogical reasoning over abstract concepts has been researched for decades [23], because of its significance in human cognition and potential applications in children assessment [4], zero-shot learning [33, 34], multimedia content understanding [28, 32, 42], etc. The recent emergence of multimodal large language models has revolutionized all aspects of daily human tasks, but due to the lack of human-like fluid reasoning capabilities, their performance in abstract thinking is far from humans [15, 34]. Analogical reasoning

**Unpublished working draft. Not for distribution.**

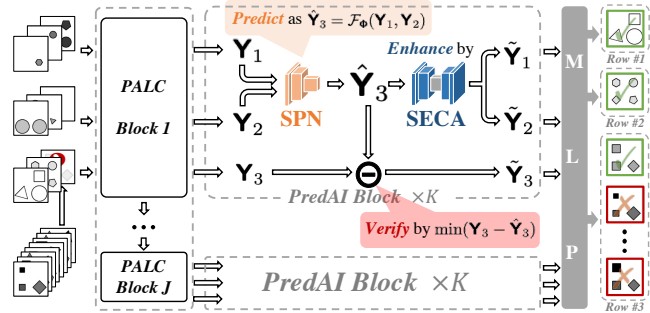

**Figure 1: Overview of the proposed HP$^2$AI framework. The proposed PredAI reasoner solves RPMs by *predicting* the 3rd entity in each row and *verifying* it against the truth, and *enhancing* important reasoning attributes in channel wise from multi-level perceptual features. This is done for the output of each of the $J$ PALC modules, which form HPALC.**

extracts high-level abstractions rationally from existing complex high-dimensional information, which enables abstract thinking beyond appearance details and comprehending unfamiliar scenarios in terms of objects, relationships and higher-order patterns [25]. It provides an effective means to assess a machine's capacity for abstraction, *i.e.*, identify common higher-order rationales by abstracting from superficial features, similar to how human reasons. Analogical reasoning can be broadly categorized based on the nature of the relations being compared, *e.g.*, categorical analogy [23], visual analogy [9] and linguistic analogy [27].

Raven's Progressive Matrices (RPM) are typical visual analogy problems whose solutions rely on an analogy over visually perceived contents [40]. RPM problems have historically been studied in cognitive science to assess human intelligence, while recently studied in computer science to improve the reasoning ability of computers [8, 36]. To minimize the impact of language barrier and culture bias, the pictorial matrices are designed with images of regular-shaped polygons, *e.g.*, triangles, circles, etc. Given a $3 \times 3$ pictorial matrix with the last one left blank, the objective is to identify the missing entry from eight candidate answers based on the visual context and inductive rules.

Solving RPM problems usually involves two main steps: visual perception and analogical reasoning. For visual perception, most RPM solvers [14, 30, 41, 43] utilize shallow neural networks to extract local visual attributes such as polygon shape, size and color, as RPM images are often constructed using simple 2D polygons or lines [1, 40]. But these shallow networks may not capture the global image layout well. Solving RPM problem requires the ability to reason about a set of relational rules inherent to global and local attributes. Deficiencies in extracting global attributes will lead to the loss of key reasoning clues, thus limit the performance

of the model. To bridge the gap, a set of hierarchically coupled Patch Attention and Local Context (PALC) blocks are proposed to extract features from multiple receptive fields. Each block in turn combines a couple of Local Context blocks built using residual blocks to capture relatively lower-level visual attributes, and a set of Patch Attention blocks to capture relatively higher-level spatial dependencies. Additionally, different from the methods that reason over object detection results [35], the proposed perceptual encoder is integrated into an end-to-end visual reasoning model without extra supervision effort in object detection.

Most existing methods often utilize object-level feature similarities to solve RPMs [2, 8, 43], *e.g.*, MRNet [2] and HCV-ARR [8] examine the pair-wise feature similarity over three rows[1] to detect recurrent object attributes. While the shared concrete object properties can be measured by *similarity*, the shared abstract relational structures can only be handled by *analogy* [6, 9]. Those models [2, 8, 43] may neglect the core nature of analogical reasoning, *i.e.*, to align relational analogies from the base and apply them to predict the target [6]. Inspired by the prediction and matching paradigm in cognitive science [6, 36], a Predictive Analogy-Inference (PredAI) framework is developed in this paper to conduct more robust analogical reasoning, in which the Analogy-Inference Blocks (AIBs) first model the abstract analogies from the base and then apply them to the target, rather than object-level similarities across different RPM images in most approaches [2, 8, 43]. As depicted in Fig. 1, the proposed AIB explicitly models the analogy by first predicting the third feature set using the first two feature sets of each row through a Shared-weight Prediction Network (SPN), and then verifying the prediction against the ground-truth. Such a formulation well emulates the human's analogical reasoning process [6], and overcomes the dilemma about "analogy" and "similarity" in RPMs.

In the proposed framework, multi-level perceptual features from different hierarchies of the perceptual encoder are jointly utilized in the reasoning module PredAI, resulting in countless potential analogical inferences embedded in these features. To discriminate essential attributes and relations from irrelevant ones, a Squeeze-and-Excitation Channel-wise Attention (SECA) is designed in the proposed PredAI, where the less relevant features and relations are suppressed at the squeeze stage and the more dominant ones are amplified with large weights at the excitation stage.

Our contributions can be summarized as follows. 1) To tackle the limited receptive fields of shallow networks in existing RPM solvers, a hierarchically coupled perceptual encoder is proposed to perceive both the local visual attributes and the global panel layout at multiple receptive fields. 2) Instead of seeking for feature similarity in existing RPM solvers, a predicting-and-verifying paradigm is designed in the proposed AIB to directly model the reasoning rules embedded in RPM questions. 3) The proposed SECA highlights the most relevant attributes and rules for solving the problem and suppresses the irrelevant ones. 4) The proposed HP$^2$AI significantly outperforms the state-of-the-art methods over 4 datasets, improving the previous best results from 98.2% to 99.3% on PGM [1], from 95.8% to 98.8% on RAVEN [40], from 96.5% to 99.4% on I-RAVEN [14] and from 97.1% to 98.6% on RAVEN-FAIR [2], respectively.

---

[1]Without losing generality, only row-wise relations are illustrated but column-wise relations are used in experiments whenever necessary.

## 2 RELATED WORK

### 2.1 Visual Reasoning

Solutions of visual reasoning can be categorized into various groups [24], including analogical (abstract) visual reasoning [1, 2, 8, 14, 24, 40], visual question answering [16, 21, 29, 37], visual commonsense reasoning [17, 19, 38, 39], mathematical visual reasoning [5, 10, 18] and many others [11, 12]. They all address the problem of reasoning for the abstract concepts present in images or the hidden patterns that govern visual entities. Visual question answering (VQA) is one of the most adapted assessments for the general reasoning ability of linking vision and language modalities, where the goal is to answer a question by referring to an associated image [21]. Analogical visual reasoning spans a variety of tasks, *e.g.*, RPMs [1, 40], visual abductive reasoning (VAR) [20], CLEVR-Matrices [25] and Bongard-LOGO problems [26]. RPMs are a typical category of analogical visual reasoning tasks, where the questions are designed in an abstract formulation using only 2D lines or shapes to minimize the impact of language barrier or culture bias. VAR [20] examines abductive reasoning ability of machine intelligence in everyday real-world situation by completing sets of visual events and inferring the hypothesis that can best explain the visual premise. Mondal *et al.* [25] developed an RPM-like problem dataset, CLEVR-Matrices, using realistically rendered 3D shapes based on the VQA dataset CLEVR [16]. The Bongard-LOGO problem [26] was developed to evaluate human-like analogy-making by interpreting a LOGO object in terms of another, with the target of discovering the concept that the positive images obey while negative images violate.

### 2.2 Solution Models for RPMs

Visual reasoning models for solving RPMs often consist of two modules: visual perception and relational reasoning. For visual perception, most solution models utilize consecutive convolution layers to extract visual features, *e.g.*, CoPINet [41] and Rel-AIR [30], and the latest PredRNet [36] all utilize residual networks [7]. Recently, MRNet [2] applies multi-scale convolutional encoders and HCV-ARR [8] adapts a mixed model of convolution blocks and transformer blocks to extract visual features from RPM images. For analogical reasoning over RPMs, early models often compare and contrast answer sets directly, *e.g.*, CoPINet [41] explicitly contrasts candidate answers and highlights the difference between options while DCNet [43] contrasts both row features and candidate options. Research shows that different levels of inductive bias are incorporated into these network structures, and thereby improve the classification performance [24, 25]. Recently, MRNet [2] deducts the correct answer by minimizing the squared Euclidean distance between row features, to identify recurring patterns. In [8], a reasoning module is designed based on the attention mechanism to dynamically learn the feature weights and highlight the important features to encode the rule representations. Most recently, PredRNet [36] applies the neural prediction errors between the selected option and the predicted answer for reasoning.

## 3 PROPOSED METHOD

### 3.1 Overview of Proposed Method

As shown in Fig. 2, the proposed HP$^2$AI consists of two modules.

**Figure 2: Block diagram of the proposed HP$^2$AI network. It mainly consists of: 1) HPALC perceptual module to visually perceive the local features at shallow stages and extract global features at deeper stages; 2) PredAI reasoning module to extract the abstract analogies by prediction/verification and to highlight critical ones for the answer.**

**Visual Perception Module.** The proposed hierarchical perceptual encoder contains a set of PALC blocks to extract the local and global image attributes at multiple receptive fields, because solving an RPM problem often requires reasoning over a set of relational rules embedded in both global high-level image semantics such as Number and Position and low-level image details such as Type, Size and Color. Each block incorporates a Local Context block modeling lower-level features through a residual network and a Patch Attention block capturing higher-level spatial layout features through a transformer network. The proposed visual encoder concurrently captures the visual cues from different receptive fields, and feeds them into the analogical reasoning modules at each receptive field for logical reasoning. As a result, the proposed method captures the image attributes of different scales and reasons over them at various receptive fields.

**Analogical Reasoning Module.** The analogical reasoner exploits the abstract analogies among features to model the underlying relations embedded in visual attributes. The analogies among visual features in RPMs are specifically designed row-wisely [1, 40]. The proposed reasoner hence divides the original $3 \times 3$ matrix along rows, and induces analogies row-wisely. At each receptive field, after extracting the visual features, a series of consecutive Predictive Analogy-Inference (PredAI) blocks are designed to capture the underlying analogies. Each PredAI block incorporates human-like analogical reasoning in two-fold: 1) Analogy Inference Block (AIB) infers the row-wise analogies embedded in the attributes of the $3 \times 3$ matrix. Specifically for the analogy on RPMs, features of the first two images are used as the input to predict the features of the third image, and the prediction function serves as the analogy. 2) Squeeze-and-Excitation Channel-wise Attention (SECA) that suppresses the irrelevant attributes and relations and highlights

the essential ones, to concentrate on the most relevant features and analogies to solve the RPM problems.

## 3.2 Visual Perception Module

Both low-level image details such as Type, Size and Color and high-level image semantics such as Number and Position are useful cues to solve RPMs [2, 8]. To extract both local context and global dependencies, a Hierarchical encoder focusing on both Local Context and Patch Attention (HPALC) is proposed. It consists of $J$ hierarchically organized PALC blocks, where features extracted from the early blocks generally focus on image fine details, whilst the latter stages are specialized to higher level features and the features become gradually sparse across channels. In traditional image encoders, features from early stages are considered too noisy for visual recognition and are usually passed into deeper stages for high-level abstraction [36]. But for RPM problems, small entity details are as important as high-level image semantics. We hence hierarchically extract features from multiple receptive fields and build PredAI block for each set of features at their respective receptive field. Furthermore, each PALC block forms a dual-branch structure to extract both local and global features, where the Local Context block perceives low-level image details of objects, and the Patch Attention block extracts global spatial semantics on images by modelling long-range dependencies through cross-patch multi-head self-attention [3, 22].

Formally, an RPM sample $\langle \mathbf{Q}, \mathbf{A} \rangle$ is composed of a $3 \times 3$ pictorial image matrix of size $H \times W$ with the last one missing, $\mathbf{Q} = \{Q_1, Q_2, \ldots, Q_8\}$ as the question set, and $\mathbf{A} = \{A_1, A_2, \ldots, A_8\}$ as the answer set. Each answer image is appended to eight question images to form one complete $3 \times 3$ RPM panel. The goal is to determine the missing image from the eight candidate answers through

visually perceiving eight panels and reasoning over the derived visual features. For example, given an RPM image $I \in \mathbb{R}^{H \times W}$ from the 3×3 panel, the Patch Attention block firstly splits $I$ into patches of size $k \times k$ to obtain the token matrix $\mathbf{P} \in \mathbb{R}^{\frac{H}{k} \cdot \frac{W}{k} \times k^2}$, where each row represents a token vector of size $k^2$ and $n_T = \frac{H}{k} \times \frac{W}{k}$ is the number of rows. Each token vector is embedded into $d_T$ dimensions through the same linear layers to obtain $E_F \in \mathbb{R}^{n_T \times d_T}$, and a positional embedding $E_P \in \mathbb{R}^{n_T \times d_T}$ is added to $E_F$ to form the input into cross-patch multi-head self-attention as $F^0 = E_F + E_P$. Next, multiple Patch Attention blocks are sequentially applied to extract the attentional features, with $F^{j-1}$ and $F^j$ as the input and the output of the $j$-th block respectively. The local self-attention (LSA) computes the self-attention relations of local patches within non-overlapped regions, whilst regional self-attention (RSA) computes the global self-attention relations between different regions of patches. As shown in Fig. 2, the extracted features for all 9 panel images are collectively denoted as $\mathbf{F}^j$. Partitioning images into patch tokens and investigating patches with shifting windows is advantageous for understanding rich combinatorial relationships such as global spatial semantics, for solving challenging problem settings such as 2x2G, 3x3G and O-IG in RAVEN datasets [8, 25, 36].

Meanwhile, the Local Context block is designed for perceiving local details of entities in RPMs. Specifically, given an image $I \in \mathbb{R}^{H \times W}$, each stage contains three successive 3×3 convolution layers with a stride of $S$ to obtain multi-scale features. The input $Z^{j-1}$ of each block is down-sampled and added to the output of the block through a shortcut to form a residual structure [7], and produce $Z^j$ as the output of the $j$-th block. Collectively, the features for the 3 × 3 RPM panel for the $j$-th Local Context block can be denoted as $\mathbf{Z}^j$. The output of the $j$-th PALC block is obtained by fusing both the output of $j$-th Local Context block and Patch Attention block,

$$\mathbf{X}^j = \mathbf{F}^j + \mathbf{Z}^j, \qquad \mathbf{X}^j \in \mathbb{R}^{9 \times D^j}, \qquad (1)$$

where $D^j$ is the number of feature channels. Hierarchical features $\{\mathbf{X}^1, \ldots, \mathbf{X}^J\}$ serve as multi-view cues for analogical reasoning.

## 3.3 Analogical Reasoning Module

For a 3×3 RPM panel, the analogies are designed row-wisely [1, 40], *i.e.*, the underlying rules can be derived from each of the first two base rows and then the same rules are applied to the third target row to infer the missing image from the options. However, it is difficult to explicitly model the rules embedded in three images of each row. As shown in Fig. 3, the proposed Analogical-Inference Block (AIB) transforms this complex and challenging problem into a predicting-and-verifying paradigm, which takes the reasoning features of the first two columns as the input, predicts those of the third column, and verifies the prediction against the features of the third column. The Shared-weight Prediction Network (SPN) is expected to capture the underlying reasoning rules across three rows. Furthermore, to suppress the irrelevant attributes and rules, and highlight the important ones, a Squeeze-and-Excitation Channel-wise Attention (SECA) mechanism is proposed to adaptively weigh the features. Finally, multiple PredAI blocks are stacked to refine rules, especially the complicated ones.

**Analogy Inference.** Most methods [2, 8, 43] attempt to refine inherent rules by maximizing the similarity over visually perceived

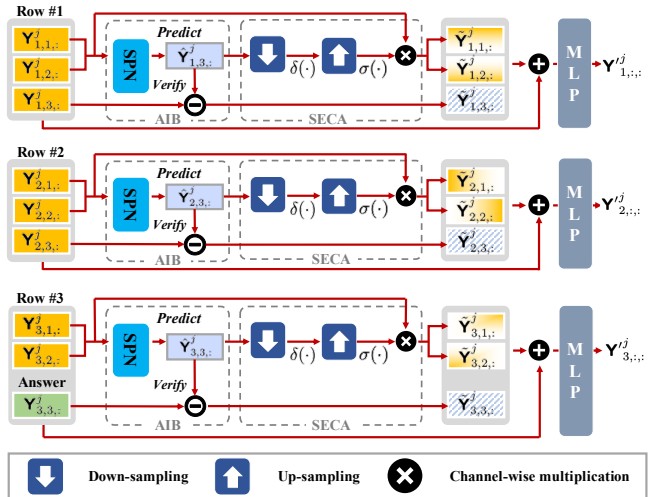

**Figure 3: Overview of the proposed PredAI reasoning module, which consists of Analogical-Inference blocks (AIBs) and Squeeze-and-Excitation Channel-wise Attention (SECA).**

row features. However, feature-level similarities do not necessarily assure similar relation-level analogies, *e.g.*, two rows of images satisfying an Arithmetic rule for Number may have different visual layouts of objects in images (Fig. 2). In contrast, the analogy inference aims at extracting analogies that fulfill relations for all three row attributes, which emulates the process of human-like high-level perceptual induction rather than purely focuses on similar visual features. In this paper, we attempt to take row-wise features perceived from the HPALC as the input, and outputs the high-level relation abstractions.

**Predicting-and-verifying Paradigm.** It is difficult to explicitly derive the underlying rules, as the perceived visual features contain multiple object attributes and hence the reasoning rules over these visual features vary significantly. Taking the feature tensors $\mathbf{X}^j$ of the $j$-th PALC block as an example, we map $\mathbf{X}^j$ to 3 × 3 layout as $\mathbf{Y}^j \in \mathbb{R}^{3 \times 3 \times D^j}$ and use $\mathbf{Y}^j$ as the reasoning features. For the $i$-th row, a naive implementation to derive the reasoning rule $\mathcal{R}_i$ is,

$$\mathcal{R}_i = \mathcal{F}_{\cdot}(\mathbf{Y}^j_{i,:,:}), \qquad (2)$$

where $\mathcal{F}_{\cdot}$ is the reasoning network and $\mathbf{Y}^j_{i,:,:}$ represent the reasoning features for the three images of the $i$-th row. To ensure that the rules are consistent across three rows, it is preferred to max $\mathcal{F}_S(\mathcal{R}_1, \mathcal{R}_2, \mathcal{R}_3)$, where $\mathcal{F}_S$ measures the similarity of three rules. Note that it is difficult to directly deduce the consistent rules across three rows as in Eqn. (2), as the analogy extraction function $\mathcal{F}_{\cdot}$ in Eqn. (2) contains huge complexity and non-linearity. To tackle this challenge, we transform the complex analogy inference problem defined in Eqn. (2) into a predicting-and-verifying problem,

$$\hat{\mathbf{Y}}^j_{:,3,:} = \mathcal{F}_{\Phi}(\mathbf{Y}^j_{:,1,:}, \mathbf{Y}^j_{:,2,:}), \qquad (3)$$

$$\tilde{\mathbf{Y}}^j_{:,3,:} = \mathbf{Y}^j_{:,3,:} - \hat{\mathbf{Y}}^j_{:,3,:}, \qquad (4)$$

where $\mathcal{F}_{\Phi}$ is a Shared-weight Prediction Network (SPN). For each row, the third entity is first predicted using the first two entities by

$\mathcal{F}_\Phi$ as in Eqn. (3), and it is verified against the ground-truth third entity as in Eqn. (4). By explicitly minimizing the prediction error $\tilde{\mathbf{Y}}^j_{:,3,:}$ in Eqn. (4), we have $\mathcal{F}_\Phi(\mathbf{Y}^j_{:,1,:}, \mathbf{Y}^j_{:,2,:}) \rightarrow \mathbf{Y}^j_{:,3,:}$, so that $\mathcal{F}_\Phi$ well models the underlying rules. In such a way, the Analogy Inference Block (AIB) extracts the analogy and applies it to predict the third entity in each row. Unlike the way in Eqn. (2) that the analogies are difficult to extract, the network parameters $\Phi$ of AIB can be explicitly refined to capture the analogies.

**Shared-weighted Prediction Network.** Many techniques have been attempted in literature to align the abstract analogies derived from visual features across all three rows, *e.g.*, Euclidean similarity [2] or attention mechanism [8]. However, these methods focus on feature-level consistency, which does not necessarily align the underlying relations. In this paper, following the predicting-and-verifying paradigm, we propose a simple but genius mechanism to restrict that the AIBs for different rows share the same weights $\Phi$. Consequently, the predicted features are derived using same weights embedded in AIBs for different rows, to ensure that the same reasoning analogy is applied across three rows.

**Squeeze-and-Excitation Channel-wise Attention.** Analogical inference through the proposed predicting-and-verifying paradigm offers an effective way to refine the network model and capture the underlying analogy, but it also poses a practical challenge, *i.e.*, between any two nonidentical features there may be countless potential inferences [6]. The high-dimensional multi-scale features implicitly represent different visual attributes perceived from the RPM images. Given one attribute dominated by one potential relation, there might be countless potential relations between three entities if derived by neural layer computations.

To tackle the challenge, a Squeeze-and-Excitation Channel-wise Attention (SECA) mechanism is integrated into the PredAI block to dynamically weigh the attributes across channels with different importance. Squeeze-and-Excitation [13] was originally designed to learn channel-wise feature dependencies to improve the network's ability to discriminate important features from less relevant ones. In this paper, the proposed SECA mechanism is designed to suppress the irrelevant features and rules that may be harmful for deriving the underlying analogies by squeezing the input feature map to a vector through global average pooling, and then the squeezed vector is excited with a set of channel-wise scaling factors that signify the importance of each channel.

Specifically, the SECA is represented by the down-sampling operation with network parameters $\mathbf{W}^j_d \in \mathbb{R}^{\frac{D^j}{r} \times D^j}$ over $\hat{\mathbf{Y}}^j_{:,3,:}$ derived from Eqn. (3) and the up-sampling operation with network parameters $\mathbf{W}^j_u \in \mathbb{R}^{D^j \times \frac{D^j}{r}}$ to obtain scalars for each row $S^j \in \mathbb{R}^{3 \times D^j}$,

$$S^j = \sigma(\mathbf{W}^j_u \delta(\mathbf{W}^j_d \mathbf{Y}^j_{:,3,:})), \qquad (5)$$

where $\delta$ denotes ReLU activation and $\sigma$ stands for the Sigmoid function. Next, the scalars $S^j$ are multiplied along the channel dimension to weigh the reasoning features as,

$$\tilde{\mathbf{Y}}^j_{:,1,d} = S^j_{:,d} \cdot \mathbf{Y}^j_{:,1,d}, \qquad \tilde{\mathbf{Y}}^j_{:,2,d} = S^j_{:,d} \cdot \mathbf{Y}^j_{:,2,d}. \qquad (6)$$

In such a way, the irrelevant features are suppressed and the important ones are highlighted. The enhanced features from Eqn. (6) and predictive differences from Eqn. (4) are combined to form $\tilde{\mathbf{Y}}^j$

and added to the original input features, and passed through multi-layer perceptrons $\mathcal{F}_{\mathsf{MLP}}$ for high-level abstraction,

$$\mathbf{Y}'^j = \mathcal{F}_{\mathsf{MLP}}(\mathbf{Y}^j + \tilde{\mathbf{Y}}^j). \qquad (7)$$

Intuitively, if the prediction error is small, $\mathbf{Y}'^j$ contains the enhanced features of the first two entities and the original features of the third entity for next stage optimization. In practice, in order to capture the complex rules, PredAI blocks $\mathcal{F}^k_{\mathsf{P}}$ are stacked as $\mathbf{Y}^j_k = \mathcal{F}^k_{\mathsf{P}}(\mathbf{Y}^j_{k-1})$, where the output from the $(k-1)$-th PredAI block is the input to the $k$-th PredAI block. Finally, the one-hot vector $\hat{p} \in \{0,1\}^8$ is derived through MLPs over all the reasoning features to determine which option is the correct answer,

$$\hat{p} = \mathcal{F}_{\mathsf{MLP}}([\mathbf{Y}^1_K; \dots; \mathbf{Y}^J_K]). \qquad (8)$$

**Discussion.** The proposed HP$^2$AI is significantly different from previous reasoners [2, 8, 43]. Its superior performance as shown in the next section is achieved mainly because: 1) We make higher-level analogy abstraction based on the perceived visual features instead of seeking object-level similarity in most previous methods [2, 8, 43]. 2) To explicitly model the underlying rules embedded in various visual attributes, we transform the problem of analogy-inference into a predicting-and-verifying problem. 3) To ensure the embedded rules are identical across rows, a Shared-weight Prediction Network is designed in the analogy-inference blocks. 4) To effectively discriminate essentials from countless possible rules associated with various attributes, the proposed SECA mechanism assigns different weights to different feature channels to signify important attributes and weaken irrelevant ones.

## 4 EXPERIMENTS

### 4.1 Experimental Settings

The proposed HP$^2$AI is systematically compared with nine state-of-the-art methods, WReN [1], CoPINet [41], DCNet [43], SRAN [14], MRNet [2], AlgebraicMR [35], HCV-ARR [8], STSN [25] and PredRNet [36] on four publicly available RPM benchmark datasets, PGM (PGM) [1], original RAVEN (O-RVN) [40], Impartial-RAVEN (I-RVN) [14] and RAVEN-FAIR (RVN-F) [2]. The key dataset statistics are summarized in Table 1. Further details of the compared methods and datasets are provided in the supplementary material.

**Table 1: RPM datasets for analogical visual reasoning.**

| Datasets | Samples | Images | Attributes | Relations |
|---|---|---|---|---|
| O-RVN [40] | 70K | 1.12M | 6 | 4 |
| I-RVN [14] | 70K | 1.12M | 6 | 4 |
| RVN-F [2] | 70K | 1.12M | 6 | 4 |
| PGM [1] | 1.42M | 22.72M | 5 | 5 |

We strictly follow the standard evaluation protocol in [1, 40]. The input image size is set to $80 \times 80$ for RAVENs and PGM. The datasets are split into training, validation, and test sets, where the validation set is utilized to determine the hyper-parameters of the model. No other forms of auxiliary supervision (*e.g.*, metadata [1]) are incorporated during training. We report the results on Neutral, Interpolation and Extrapolation regimes for the PGM dataset,

**Table 2: Comparison with state-of-the-art on the original RAVEN dataset [40]. † indicates that the original method is based on contrasting over candidate answers. The proposed method significantly outperforms all the compared methods over 7 configurations consistently.**

| Methods | Accuracy (%) on Different Configurations | | | | | | | |
|---|---|---|---|---|---|---|---|---|
| | Avg. | Center | 2x2G | 3x3G | L-R | U-D | O-IC | O-IG |
| †CoPINet (NIPS'19) [41] | 91.4 | 95.1 | 77.5 | 78.9 | 99.1 | 99.7 | 98.5 | 91.4 |
| †DCNet (ICLR'21) [43] | 93.6 | 97.8 | 81.7 | 86.7 | 99.8 | 99.8 | 99.0 | 91.5 |
| †HCV-ARR (AAAI'23) [8] | 96.0 | 99.4 | 86.9 | 89.1 | 99.9 | 99.9 | 99.8 | 96.8 |
| †MRNet (CVPR'21) [2] | 96.6 | 99.9 | 97.8 | 91.2 | 99.7 | 99.7 | 99.6 | 87.7 |
| SRAN (AAAI'21) [14] | 46.2 | 49.0 | 45.4 | 52.8 | 42.4 | 36.0 | 49.1 | 48.8 |
| MRNet (CVPR'21) [2] | 84.0 | 98.7 | 72.5 | 52.3 | 99.4 | 99.2 | 99.6 | 66.3 |
| HCV-ARR (AAAI'23) [8] | 87.3 | 99.8 | 71.4 | 65.9 | 99.9 | 99.8 | 98.0 | 76.2 |
| AlgebraicMR (CVPR'23) [35] | 92.9 | 98.8 | 91.9 | 93.1 | 99.2 | 99.1 | 98.2 | 70.1 |
| PredRNet (ICML'23) [36] | 95.8 | 99.8 | 95.1 | 87.6 | 99.2 | 99.4 | 99.9 | 89.4 |
| HP$^2$AI (*Ours*) | **98.8** | **100.0** | **98.8** | **95.3** | **99.9** | **99.8** | **99.9** | **98.0** |

**Table 3: Comparison with state-of-the-art models on the I-RAVEN/RAVEN-FAIR datasets [2, 14]. The proposed method significantly outperforms all the compared methods over all configurations consistently on both datasets.**

| Methods | Accuracy (%) on Different Configurations | | | | | | | |
|---|---|---|---|---|---|---|---|---|
| | Avg. | Center | 2x2G | 3x3G | L-R | U-D | O-IC | O-IG |
| DCNet (ICLR'21) [43] | 46.6/57.0 | 56.2/57.2 | 32.7/48.4 | 32.9/58.2 | 54.7/57.5 | 53.9/59.4 | 55.9/62.0 | 39.8/56.2 |
| SRAN (AAAI'21) [14] | 60.8/76.7 | 78.2/87.4 | 50.1/60.4 | 42.4/62.8 | 70.1/86.5 | 70.3/86.7 | 68.2/77.5 | 46.3/75.9 |
| MRNet (CVPR'21) [2] | 81.0/86.8 | 99.6/97.0 | 63.4/72.7 | 59.2/69.5 | 98.7/98.7 | 98.3/98.9 | 95.7/97.6 | 51.9/73.3 |
| AlgebraicMR (CVPR'23) [35] | 93.2/94.3 | 99.5/99.8 | 89.6/93.2 | 89.7/88.0 | 99.7/99.8 | 99.5/99.8 | 99.6/99.9 | 74.7/79.6 |
| HCV-ARR (AAAI'23) [8] | 93.9/95.4 | 99.9/99.8 | 96.2/92.9 | 75.5/87.9 | 99.4/99.8 | 99.6/99.6 | 99.5/99.7 | 87.3/88.5 |
| STSN (ICLR'23) [25] | 95.7/95.4 | 98.6/98.4 | 96.2/95.8 | 88.8/87.3 | 98.0/98.3 | 98.8/98.1 | 97.8/98.2 | 92.0/92.2 |
| PredRNet (ICML'23) [36] | 96.5/97.1 | 99.9/99.8 | 97.8/97.3 | 91.2/92.6 | 99.7/99.7 | 99.7/99.5 | 99.6/99.7 | 87.7/91.2 |
| HP$^2$AI (*Ours*) | **99.4/98.6** | **100.0/100.0** | **99.9/99.4** | **97.4/96.9** | **99.9/99.9** | **100.0/99.9** | **100.0/99.7** | **98.8/94.2** |

similar to many previous models [2, 25, 36, 43]. For RAVENs, we adhere to the settings described in their original papers for a fair comparison. Adam optimizer is used with a learning rate of 1e-3 and a batch size of 128. For the RAVENs, the weight decay is set at 1e-5, while for the PGM, it is 1e-7 due to their larger quantity.

## 4.2 Comparisons with State-of-the-Art Models

**Original RAVEN Dataset** [40] contains a loophole that the correct answer can be derived by simply aggregating the most common properties from the answer options, without examining the question at all, as identified in [2, 14]. We hence conduct evaluations without contrasting over options, following most previous models [2, 8, 36], and the results are summarized in Table 2. From Table 2, we can see that some methods utilizing the contrast information achieve high accuracy, while their performance drops if the contrast information is not utilized, *e.g.*, MRNet [2] and HCV-ARR [8]. In contrast, the proposed method reaches the mean accuracy of 98.8% without any auxiliary contrast information over options, which outperforms the second-best method, PredRNet, by 3.0%, and it is even superior to the methods using the contrast information. The proposed method significantly outperforms all the compared

methods for all configurations consistently. Specifically, it achieves significant improvements on complicated configurations where high-level spatial positions are considered, *e.g.*, 2x2G, 3x3G and O-IG, which demonstrates the effectiveness of the Patch Attention blocks in visual perception for global feature extraction. The proposed PredAI module effectively models the underlying relations following the designed analogy inference mechanism, which yields one shared rationale for each attribute across rows, and discovers a wide range of reasoning rationales for better reasoning.

**I-RAVEN Dataset** [14] has been developed to eliminate the loophole in the original RAVEN dataset [40]. Table 3 summarizes the reasoning accuracy on the I-RAVEN dataset [14]. The proposed HP$^2$AI significantly outperforms the state-of-the-art methods on the I-RAVEN dataset, achieving an average reasoning accuracy of 99.4%, with 100.0% perfect or 99.9% near-perfect over five out of the seven problem configurations. Compared with the second best method, PredRNet [36], the superiority is 2.9% on average, with massive improvements on those complicated configurations, *e.g.*, 6.2% on 3x3G and 11.1% on O-IG. All these results are achieved by hierarchically perceiving RPM images in multiple receptive fields and the effectiveness of the proposed PredAI blocks in discovering high-level abstractions of embedded analogies.

**RAVEN-FAIR Dataset** [2] has been developed to avoid the loophole in the original RAVEN dataset by generating option images with more randomness. The results of comparison to state-of-the-art methods are summarized in Table 3. The proposed HP$^2$AI achieves an average reasoning accuracy of 98.6% on the RAVEN-FAIR dataset [2], which consistently outperforms all the compared methods, except that for O-IC configuration it achieves the same accuracy of 99.7% as the second best method. Compared with the second best method PredRNet [36], the performance gains on challenging configurations are most significant, *e.g.*, 2.1% on 2x2G, 4.3% on 3x3G and 3.0% on O-IG.

In summary, the proposed method shows significant performance gains on all three RAVEN datasets consistently, particularly on complex 2x2G, 3x3G and O-IG configurations involving spatial relations, which confirm the positive contribution of the proposed HPALC and PredAI modules.

**Table 4: Test accuracy (%) of three different regimes on PGM.**

| Methods | Accuracy (%) on Different Regimes | | |
|---|---|---|---|
| | Neutral | Interpolation | Extrapolation |
| CoPINet [41] | 56.4 | 51.2 | 16.4 |
| WReN [1] | 62.6 | 64.4 | 17.2 |
| DCNet [43] | 68.6 | 59.7 | 17.8 |
| SRAN [14] | 71.3 | 60.1 | 18.4 |
| MRNet [2] | 93.4 | 68.1 | 19.2 |
| PredRNet [36] | 97.4 | 70.5 | 19.7 |
| STSN [25] | 98.2 | 78.5 | 20.4 |
| **HP$^2$AI (*Ours*)** | **99.3** | **80.0** | **22.6** |

**PGM Dataset** [1] has been widely used for evaluating the capability of abstract reasoning [2, 14, 25, 36, 43]. The proposed method is compared with state-of-the-art models on the PGM dataset in Table 4. Some methods such as CoPINet [41] and DCNet [43] that perform satisfactorily on the RAVENs may not perform well on the PGM dataset. This is mainly because these models adopt RAVEN-specific inductive biases rather than investigate the core of analogical reasoning, and hence have poor generalization abilities. The proposed method achieves 99.3% reasoning accuracy, which significantly outperforms the 98.2% accuracy of the previous best method STSN [25] and the 97.4% accuracy of the second best method PredRNet [36] on the Neutral regime. On the other two OOD regimes, the proposed method also demonstrates good generalization ability by improving the previous best method STSN [25] by 1.5% and 2.2%, respectively. The PGM dataset is 20 times larger than RAVENs, and the rules are applied either in rows or in columns. Despite these additional constraints, our method is able to obtain a significant gain over a rather high accuracy of 98.2%, thanks to the hierarchical design in HPALC for visual perception from multiple receptive fields and the predicting-and-verifying design in the PredAI reasoning blocks.

## 4.3 Ablation Studies

**Ablation Studies of Major Components.** We ablate the two main modules of the proposed HP$^2$AI on three RAVEN-style datasets [2,

14, 40]. We select three representative models such as MRNet [2], HCV-ARR [8] and PredRNet [36] as the baselines. We disentangle the perception module and reasoning module for the baselines, and substitute the respective module by the proposed one. Specifically, we ablate the proposed HP$^2$AI using perception modules Multi-Scale Encoder (MSE) from MRNet [2], Hierarchical ConViT (HCV) from HCV-ARR [8] and 4-Block ResNet (RN-4B) from PredRNet [36]. We also ablate the proposed HP$^2$AI using reasoning modules Relation Module with Pattern Module (RM+PM) from MRNet [2], Attention-based Relation Reasoner (ARR) from HCV-ARR [8] and Predictive Reasoning Block (PRB) from PredRNet [36]. The results are summarized in Table 5. We can observe the consistent performance gain brought by the two proposed modules over the baselines. The average gains over three datasets brought by the proposed HPALC and the proposed PredAI are 1.3% and 1.4% compared with RN-4B and PRB from PredRNet [36], 1.0% and 2.4% compared with HCV and ARR from HCV-ARR [8], and 1.0% and 2.5% compared with MSE and PM+RM from MRNet [2], respectively. Furthermore, the SECA block of the PredAI module brings a substantial average gain of 0.4%. The ablation results demonstrate the effectiveness of the proposed HPALC in visual perception and the proposed PredAI in analogical reasoning.

**Table 5: Ablation studies of the two major components of the proposed HP$^2$AI on three RAVEN-style datasets [2, 14, 40].**

| Major Module | | Accuracy (%) on Datasets | | | |
|---|---|---|---|---|---|
| Visual | Reasoning | O-RVN | I-RVN | RVN-F | **Avg.** |
| MSE [2] | RM+PM [2] | 84.0 | 81.0 | 86.8 | 83.9 |
| HCV [8] | ARR [8] | 87.3 | 93.9 | 95.4 | 92.2 |
| RN-4B [36] | PRB [36] | 95.8 | 96.5 | 97.1 | 96.5 |
| MSE [2] | *AIB+SECA* | 86.2 | 84.0 | 88.9 | 86.4 |
| HCV [8] | *AIB+SECA* | 90.9 | 95.8 | 97.1 | 94.6 |
| RN-4B [36] | *AIB+SECA* | 97.7 | 98.1 | 97.8 | 97.9 |
| *HPALC* | RM+PM [2] | 85.1 | 81.8 | 87.7 | 84.9 |
| *HPALC* | ARR [8] | 88.5 | 95.2 | 96.0 | 93.2 |
| *HPALC* | PRB [36] | 97.2 | 98.3 | 98.0 | 97.8 |
| *HPALC* | *AIB* | 98.6 | 98.6 | 98.3 | 98.5 |
| *HPALC* | *AIB+SECA* | **98.8** | **99.4** | **98.6** | **98.9** |

**Visualization of HPALC Features.** The design of hierarchical visual perception in the proposed PredAI is essential and fundamental for achieving high reasoning performance, where shallower layers focus more on local features and deeper layers focus more on global features. We conduct the t-SNE analysis [31] for HPALC features extracted at different depths, as shown in Fig. 4a. It can be observed that in shallower layers, problem configurations that contain higher-level 'Out-In' spatial combinations are unable to discriminate, *i.e.*, O-IC and O-IG. When the network goes deeper as shown in Fig. 4b, clear improvements can be noticed about discrimination power over O-IC and O-IG. Finally, the perception module can perfectly distinguish all 7 configurations with clear boundaries in Fig. 4c. The t-SNE plots validate our analysis that shallower layers mainly focus on capturing local attributes while deeper layers mainly focus on capturing global attributes.

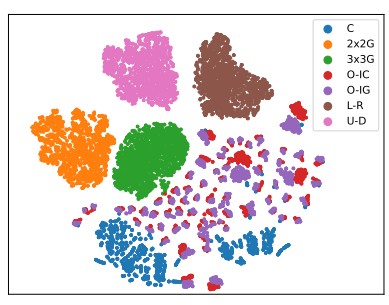

(a) Features from the 1st stage of HPALC.

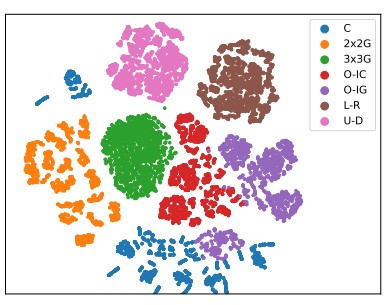

(b) Features from the 2nd stage of HPALC.

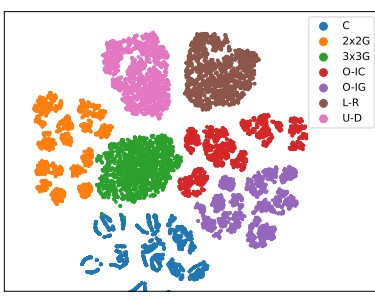

(c) Features from the 3rd stage of HPALC.

Figure 4: The t-SNE analysis for the features extracted from different stages of HPALC.

Table 6: Ablation studies of different hierarchical stages $J$ in HP$^2$AI and different number of PredAI blocks for the proposed HP$^2$AI on all three RAVEN datasets [2, 14, 40].

| | O-RVN | I-RVN | RVN-F | Avg. |
|---|---|---|---|---|
| **Number of hierarchical stages $J$ in HP$^2$AI** | | | | |
| $J = 1$ | 87.1 | 88.9 | 91.4 | 89.1 |
| $J = 2$ | 92.4 | 93.6 | 96.4 | 94.1 |
| $J = 3$ | 98.8 | 99.4 | 98.6 | 98.9 |
| $J = 4$ | 98.6 | 99.1 | 98.5 | 98.7 |
| **Number of PredAI blocks $K$** | | | | |
| $K = 1$ | 98.1 | 98.6 | 98.3 | 98.3 |
| $K = 2$ | 98.4 | 99.1 | 98.3 | 98.6 |
| $K = 3$ | 98.8 | 99.4 | 98.6 | 98.9 |
| $K = 4$ | 98.0 | 99.0 | 98.2 | 98.4 |

**Impact of Number of ReST Blocks.** At different depths of the proposed method, the features encoded by ReST is fed into a set of PredAI blocks for analogical reasoning. This design not only perceives multi-scale visual features, but also conducts multi-level reasoning in a hierarchical manner. We ablate the number of hierarchical stages $J$ on three RAVEN-style datasets [2, 14, 40] and summarize the results in Table. 6. At shallower stages, the model captures low-level feature attributes and reasons over these local attributes only using abstract analogies, but rarely captures high-level spatial information at shallow stages. The accuracy for $J = 1$ is hence low for all three datasets. When $J$ increases, the network goes deeper and the reasoning accuracy improves with deeper stages, e.g., the overall accuracy improves by 5.0% when $J$ increases from 1 to 2, and further improves by 4.8% when $J$ increases from 2 to 3. However when $J = 4$, the feature maps at Stage 4 are too small to capture discriminative visual information for analogical reasoning, which imposes a slight negative impact on the reasoning accuracy.
**Impact of Number of PredAI Blocks.** To evaluate the impact of the number of PredAI blocks, an ablation study is carried out on three RAVEN-style datasets and the results are summarized in Table. 6. It can be observed that the reasoning accuracy increases with the increasing $K$ on all three datasets, and reaches the maximum at $K = 3$. Our hypothesis is that the early PredAI blocks may not well capture the underlying rules due to the light model of SPN,

and hence result in some inaccurate predictions. As $K$ increases, a deeper network can better capture the complicated rules. When $K = 4$, the performance drops slightly, probably due to over-fitting.
**Visualization of Verification Errors.** Fig. 5 plots verification errors shown in Eqn. (4) between the prediction and ground-truth on the train and validation sets for different number of PredAI blocks $K$ and hierarchical stages $J$. It shows that the verification error decreases as training proceeds, though it is not explicitly used as a loss term.

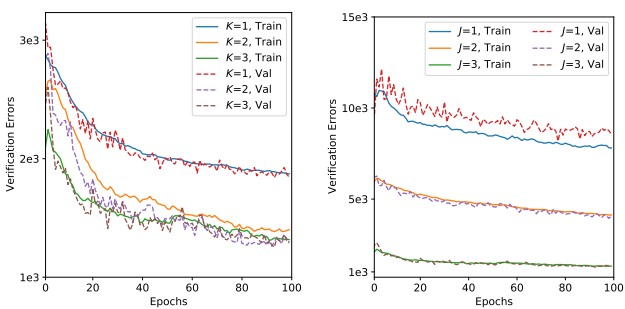

(a) Errors for different $K$ at $J = 3$. (b) Errors for different $J$ at $K = 3$.

Figure 5: Visualization of verification errors on I-RVN [14].

## 5 CONCLUSION

In this work, a novel HP$^2$AI network is proposed to solve Raven's Progressive Matrices, one of the most frequently-used assessments for human's reasoning capabilities. The proposed HPALC module simultaneously extracts the fine image details and global image semantics through the hierarchical Patch Attention and Local Context blocks across multi-level receptive fields. Instead of detecting object similarity, the proposed PredAI module models and exploits the inherent analogical reasoning rules embedded in RPM questions. We also propose the Squeeze-and-Excitation Channel-wise Attention to discriminate the importance of feature channels and hence to control their contribution to the final decision. Extensive comparisons with nine state-of-the-art models on four benchmark datasets demonstrate that the proposed HP$^2$AI significantly outperforms all state-of-the-art models consistently in all configurations and on all four datasets.

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
