# OpenReview forum: "Hierarchical Perceptual and Predictive Analogy-Inference Network for Abstract Visual Reasoning"
_acmmm.org/ACMMM/2024/Conference — MM2024 Poster_

### Official Review · Reviewer_QpbP · 2024-05-27

**Rating:** 2
**Confidence:** 1

**Summary:**

The paper describes a model to solve Raven’s Progressive Matrices (RPMs) problems, in which the model must pick the correct image out of a pool of answers that completes the third row of a 3x3 image matrix, following the rules that can be deducted from the first two rows.

The main contributions seem architectural and specifically tailored to improve performance on RPM-related datasets and comprise custom modules for feature extraction (HPALC) and reasoning (PredAI).

**Strengths:**

Results of experiments show an improved performance on multiple datasets.

**Limitations:**

The main limitation, in my opinion, is that the work is out of scope for the Multimedia conference. Although the MM community has thrived on progress in neural processing pipelines, the work is not multimodal and does not tackle multimedia. Indeed, the state-of-the-art methods the authors compared with have been presented to neural processing/computer vision/general AI/ML venues.

Moreover, the presentation of the proposed architecture should be made clearer, as there are many modules and acronyms that the reader should keep in mind, each with its specific function and peculiarities and most containing custom modifications. This makes it very difficult to discern which component is a reused one or an incremental change of an existing one (e.g., HPALC seems to me like a two-branch parallel ViT-ResNet architecture) and which is an original formulation. Some proposed modules in the architecture are justified in the text but not rigorously tested; e.g., the authors introduced SECA to tackle a particular challenge discussed in L484-493. How did the authors quantitatively measure that the introduced mechanism solved or mitigated that hypothesized issue?

For this reason, the overall architecture feels like a bag of tricks guided by performance and inductive biases rather than hypothesis testing. Thus, I did not see clear takeaway messages in this study. I perceived most of the design choices justified as “because it works" without taking the opportunity to analyze “why it works”.

**Suitability:**

1

---

### Official Review · Reviewer_1xhc · 2024-05-28

**Rating:** 4
**Confidence:** 1

**Summary:**

The paper introduces HP2AI, a Hierarchical Perception and Predictive Analogy-Inference network designed for abstract visual reasoning tasks like Raven’s Progressive Matrices (RPMs). HP2AI uses a perceptual encoder with Patch Attention and Local Context blocks to capture local and global image features, and a Predictive Analogy-Inference module to model analogical reasoning rules. Experiments on four RPM benchmark datasets show that HP2AI outperforms current state-of-the-art methods in accuracy.

**Strengths:**

1. HP2AI consistently outperforms existing models across multiple RPM benchmark datasets, indicating its strong reasoning capabilities.
2. The Predictive Analogy-Inference module enhances the model’s ability to understand and apply analogical reasoning, moving beyond simple feature similarity.
3. The model demonstrates strong performance in different test regimes, including neutral, interpolation, and extrapolation settings.

**Limitations:**

1. While the model shows strong performance on the tested RPM datasets, its scalability to larger and more diverse datasets is not thoroughly explored. Real-world applications often involve more varied and complex data, and it is unclear if the HP2AI model can maintain its performance in these scenarios without significant modifications.
2. The paper provides limited details on the implementation aspects, such as hyperparameter settings, training durations, and computational requirements.
3. The strong performance of HP2AI on the benchmark datasets raises concerns about potential overfitting. The model might be overfitting to the specific characteristics of the RPM datasets rather than learning generalizable reasoning skills. This is particularly concerning given the model's high accuracy scores, which might not translate to unseen data or different types of visual reasoning tasks.
4. The effectiveness of the HP2AI model is heavily dependent on the quality and structure of the training data. If the training data is not well-curated or if it lacks diversity, the model's performance could degrade significantly.

**Suitability:**

1

---

### Official Review · Reviewer_GaNC · 2024-05-28

**Rating:** 4
**Confidence:** 2

**Summary:**

The author proposed $HP^2AI$, a method that targets analogical visual reasoning problems. The proposed method contains two different modules: the visual perception module and the analogical reasoning module. Specifically, the visual perception module can not only capture low-level features like type, size, and color but also capture high-level features like position, and number. The analogical reasoning module uses the first two images to predict the third image and optimize the model. Experiment results show the effectiveness of the proposed method.

**Strengths:**

1. The paper is well-organized and easy to follow.
2. The paper provides comprehensive experiment results to prove the effectiveness of the proposed method.
3. The idea of the paper is straightforward and follows the intuition of humans.

**Limitations:**

1. According to the author's claim, at shallower stages, the visual perception module should focus on low-level features like shape, and color. However, as shown in Figure 4, the t-SNE results show that the features of HPALC are already unique for different configurations.  In my understanding, the low-level feature should be similar in these configurations, why the t-SNE results are unique in these configurations?
2. In the HPALC, in my understanding, the author used the addition operation to fuse the $F^j$ and $Z^j$, could you use the concatenate operation to fuse these two features and provide the quantitative results?
3. As shown in Figure 5, looks like the verification error keeps decreasing for 100 epochs, whether the model can get higher accuracy if keep training?

**Suitability:**

2

---

### Meta-Review · Area_Chair_vjq2 · 2024-06-28

**Recommendation:** Accept (Poster)
**Confidence:** 3

**Metareview:**

This paper obtained two borderline accepts and one borderline reject in the final evaluation. However, the borderline reject reviewer is not against the acceptance of this paper given the improvements in the presentation of hypotheses and solutions can be made in the camera-ready version.

The AC would like to accept this paper for its contribution to a novel network of abstract visual reasoning and its strong performance shown in the experiments. The AC urges the authors to make more efforts on editing the paper for a clearer presentation of hypotheses and solutions as suggested by the third reviewer.